# Snowmelt-Driven Seasonal Infiltration and Flow in the Upper Critical Zone, Niwot Ridge (Colorado), USA

**David P. Dethier** [1,*] **, Noah Williams** [1] **and Jordan F. Fields** [1,2]

1   Department of Geosciences, Williams College, Williamstown, MA 01267, USA;
    noah.north.williams@gmail.com (N.W.); jordan.f.fields.gr@dartmouth.edu (J.F.F.)
2   Department of Earth Sciences, Dartmouth College, Hanover, NH 03755, USA
*   Correspondence: ddethier@williams.edu

**Abstract:** The hydrology of alpine and subalpine areas in the Colorado Front Range (USA) is evolving, driven by warming and by the alteration of precipitation patterns, the timing of snowmelt, and other components of the hydrologic budget. Field measurements of soil hydraulic conductivity and moisture along 30-m transects ($n$ = 13) of representative soils developed in surficial deposits and falling head slug tests of shallow groundwater in till demonstrate that hydraulic conductivity in the soil is comparable to hydraulic conductivity values in the shallow aquifer. Soil hydraulic conductivity values were variable (medians ranged from $5.6 \times 10^{-7}$ to $4.96 \times 10^{-5}$ m s$^{-1}$) and increased in alpine areas underlain by periglacial deposits. Hydraulic conductivities measured by a modified Hvorslev technique in test wells ranged from $4.86 \times 10^{-7}$ to $1.77 \times 10^{-4}$ m s$^{-1}$ in subalpine till. The results suggest a gradient from higher hydraulic conductivity in alpine zones, where short travel paths through periglacial deposits support ephemeral streams and wetlands, to lower hydraulic conductivity in the till-mantled subalpine zone. In drier downstream areas, streambed infiltration contributes substantially to near-channel groundwater. As summer temperatures and evapotranspiration (ET) increase and snowmelt occur earlier, alpine soils are likely to become more vulnerable to drought, and groundwater levels in the critical zone may lower, affecting the connectivity between late-melting snow, meltwater streams, and the areas they affect downstream.

**Keywords:** infiltration capacity; saturated hydraulic conductivity; periglacial; evapotranspiration; groundwater; critical zone

## 1. Introduction

Over 16% of the world's population uses water that originates from glaciers and seasonal snowpacks [1]. Snowmelt-dominated mountain catchments are biologically sensitive areas that help supply water to much of the population of the western United States and Canada and other upland areas of the globe [2–6]. Increases in surface air temperatures alter the seasonality of the water supply in these areas and are likely to result in less precipitation falling as snow, earlier onset of spring snowmelt, earlier peak runoff, and, therefore a decreased late-season water supply when demand is greatest [1,7–10]. In Colorado, the 3100 to 3700 m elevation zone of subalpine and alpine areas generates most snowmelt runoff [11] and soil hydraulic conductivity is an important control for the suite of transport processes and storage elements grouped as mass balance (Equation (1)) in the critical zone of these upland catchments. The critical zone includes the soil and underlying shallow subsurface down to unweathered material, the "critical" area for hydrologic and biogeochemical processes that sustain life [12]. Seasonal infiltration of snowmelt into the soil ($I$) and through the shallow critical zone to recharge groundwater ($R$) links hydrologic process and storage (Figure 1), modified by evaporation (E) and evapotranspiration (ET) losses [13], and by the temporally and spatially variable storage of water in winter snowpacks and in the several meters of regolith that covers bedrock: Despite long-term hydrometeorological

studies, uncertainties persist about hydrologic connectivity and in the water budget at Niwot Ridge, Colorado (Figure 2) [13–15] and similar alpine and subalpine settings in mountainous landscapes.

$$P - E - ET - Q = \Delta S \tag{1}$$

where P = precipitation (mm)
E = evaporation and sublimation (mm)
ET = evapotranspiration (mm)
Q = surface (**sw**) and groundwater (**gw**) flow
out of the catchment (mm) and
ΔS = change in water stored in snowpack(**s**),
regolith (**rw**) and bedrock (**gw**) (mm)

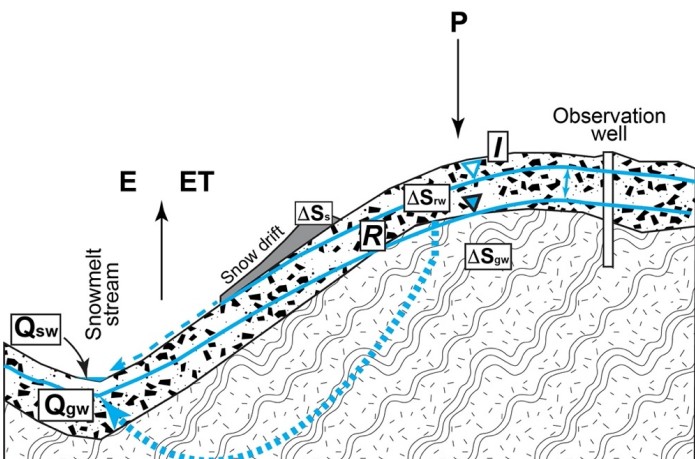

**Figure 1.** Conceptual model showing water flow and storage in the alpine zone. Flow includes precipitation (P), evaporation (E) and evapotranspiration (ET), and discharge (Q) as surface water (**sw**) and groundwater (**gw**). Storage (**S**) includes snow (**s**), regolith water (**rw**) and groundwater (**gw**). Infiltration (*I*) is water that enters the soil, and recharge (*R*) is flow that reaches the water table in the alpine and subalpine critical zone where several meters of regolith cover bedrock. Seasonal positions of the water table and a groundwater flow line are shown in blue. Infiltration through the streambed is not portrayed.

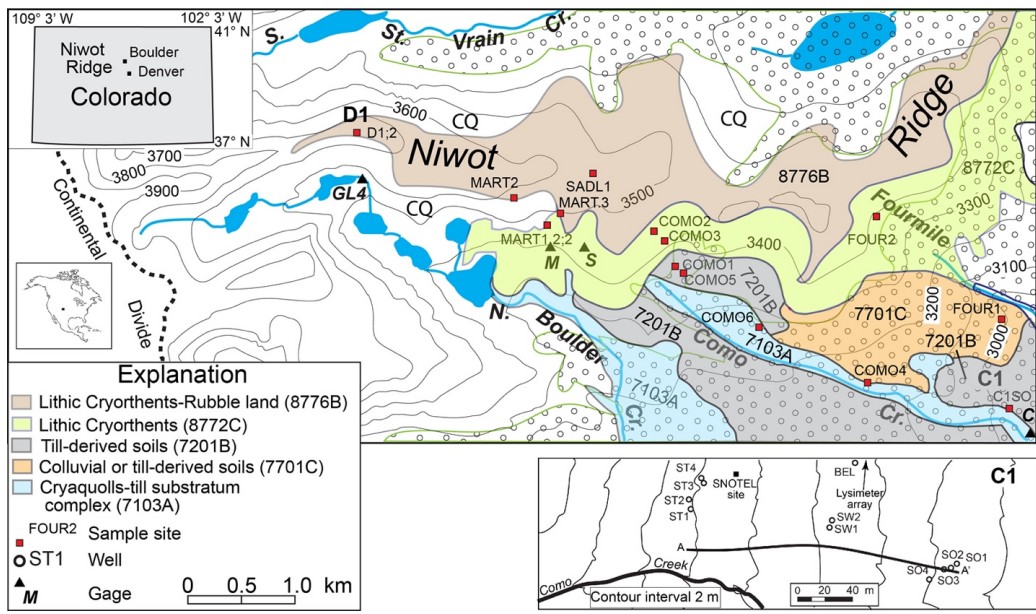

**Figure 2.** Map showing location of Colorado, Niwot Ridge and sites noted in the text. Sampled soils

were mapped by the NRCS [16] and include areas of CQ (cirque deposits). Stipple shows extent of forested area. Inset shows detail of sample sites at C1 and location of Como Creek and cross-section A–A'. Several flood channels are located between Como Creek and the cross-section. Soil mapping not shown south and west of N. Boulder Creek.

Changes in seasonal snowpack and snowmelt timing produce variable short-term and yearly responses in surface-groundwater interactions and channel flow in the critical zone [17–19]. Infiltration is strongly seasonal, and the transport and storage of water in alpine and subalpine basins is highly sensitive to soil hydraulic conductivity. This study extends previous hydrologic studies on Niwot Ridge, one of the most carefully studied alpine sites in North America, to a semi-distributed sampling of soils and geologic materials in the area and adds to the hydrogeologic database for mountainous alpine and subalpine environments. Our principal objectives are: (1) establishing the probable range of soil hydraulic conductivity values, particularly for periglacial deposits, for use in hydrologic models, (2) measuring hydraulic conductivity values for lower-elevation subalpine till derived from gneiss and granite; and (3) suggesting how seasonal changes in connectivity affect water flow in alpine and lower areas.

## 2. Setting

Niwot Ridge is located 35 km west of Boulder, Colorado, USA [16] and slopes east ~9 km from the Continental Divide as a broad alpine and subalpine interfluve between the St. Vrain and N. Boulder Creek valleys (Figure 2). The east-trending, glacially-sculpted valleys support tiny glaciers at present and were partially filled with ice during the local last glacial maximum (LLGM), the Pinedale glaciation of ~32–12 ky BP. Most of Niwot Ridge remained above the glacial limit but supported processes that produced periglacial deposits from shattering and transport of the high-grade metamorphic (gneiss and schist) and granitic bedrock of the ridge. Till is confined to the deep valleys 100 to > 250 m below the ridges, typical of many ranges in the western USA and Europe where glacial ice was thin or absent in upland areas between valleys. The soils are mainly 30–60 cm thick Inceptisols and Alfisols in areas underlain by periglacial deposits and 60 to >100 cm deep on Pinedale and older tills [16,20]. Most of the Niwot Ridge crest is mapped as a Lithic Cryorthents-Rubble land complex (Minimal Snow Cover Soil Complex of [20] and includes extensive areas of openwork boulders and patterned ground active before the 21st century.

The Niwot Ridge area is part of the Long-Term Ecological Research Network (National Science Foundation, NWT LTER), a Biosphere Reserve (United Nations Educational, Scientific and Cultural Organization, UNESCO), an Experimental Ecology Reserve (USDA Forest Service), the Boulder Creek Critical Zone Observatory and a core terrestrial NEON site https://www.neonscience.org/field-sites/field-sites-map/NIWO accessed on 1 July 2022). The weather and climate have been monitored on Niwot Ridge since 1952 by the Mountain Climate Program of the University of Colorado.

### 2.1. Geophysical and Geologic Observations of the Shallow Subsurface, Niwot Ridge

Geophysical investigations (resistivity, seismic refraction, and ground-penetrating radar) of the shallow subsurface, combined with geologic analysis of exposures and borehole information, define the materials and geologic layering of the upper 10 m of the critical zone on Niwot Ridge and adjacent areas [21]. The soils have developed on clast-rich periglacial deposits and, at lower elevations, on till. Fractured igneous and metamorphic bedrock is present at depths of 3 to >10 m and crops out at the surface in <5% of the study area. Permafrost reported in the 1970s appears to have melted at most elevations below ~3500 m [22–24]. The material on the slopes is layered on a decimeter to meter scale and locally includes several generations of solifluction deposits (Figure 3) and areas of openwork boulders beneath thin soils. Coarse layers provide storage and subsurface pathways to surface channels e.g., [25].

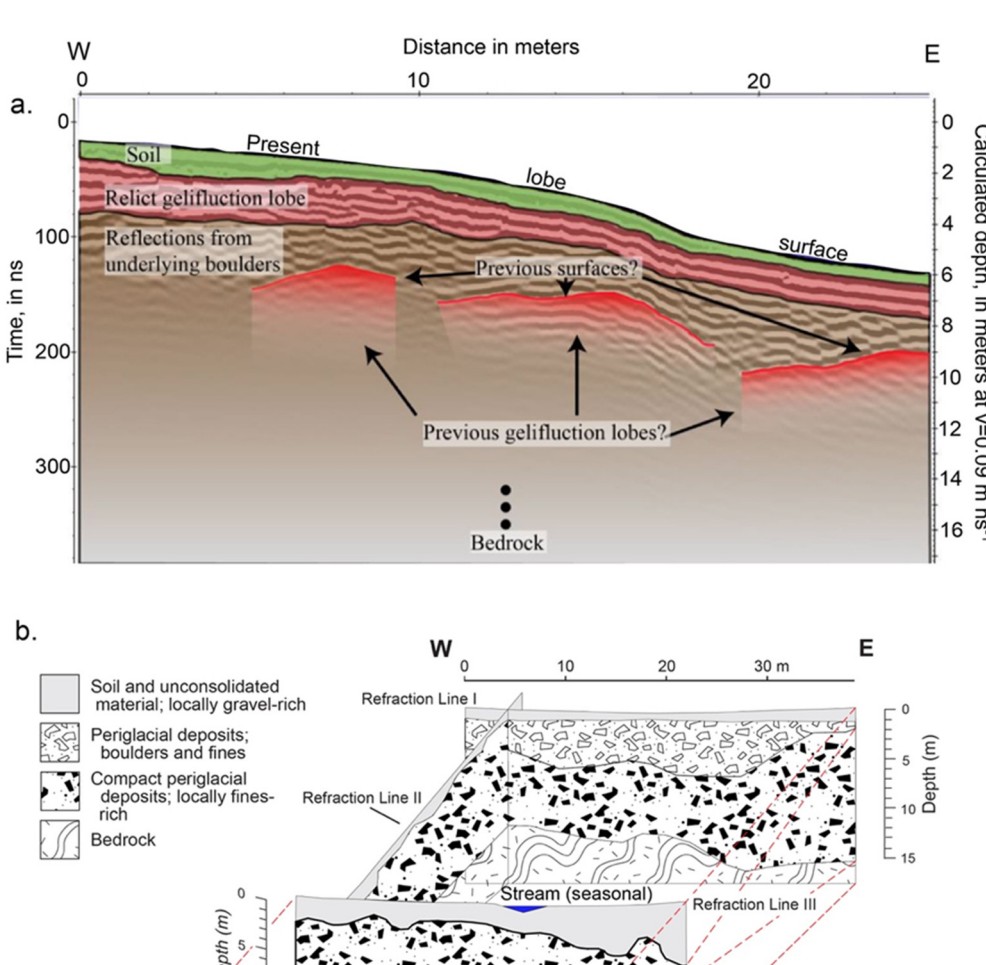

**Figure 3.** Subsurface model of Niwot Ridge from geophysical measurements. (**a**) Geologic interpretation of ground-penetrating radar line (modified from [26], p.116, Fig. 68b) from 100 m SW of SADL 1 (see Figure 2). (Adapted with permission from Gabe Lewis, Copyright 2013 to Gabe Lewis). (**b**) Fence diagram showing inferred layering of the upper critical zone near upper Como Creek, based on shallow seismic reflectometry (SSR) measurements (Reprinted/adapted with permission from ref. [21], p. 522, Figure 3. Copyright 2008 Arctic, Antarctic, and Alpine Research).

## 2.2. Niwot Ridge Climate and Hydrology

The climate in the alpine and subalpine areas of Niwot Ridge is cold and continental. High winter winds transport low-density, fresh snow, increasing the effective precipitation in protected and lee areas [27] and producing soils that have properties different than those in adjacent, windswept areas; the tree line is ~3400 m. The local soil properties reflect differences in snowpack thickness and duration in drifts over the melt season; upland catchments also receive significant late-season moisture as spatially variable convective rainfall, which accounts for >20% of the annual precipitation. Recent data suggest that the climate of Niwot Ridge is warming [13,24,28,29]. The mean winter air temperatures have remained stable, while the mean summer and autumn air temperatures have risen almost 2 °C at D1 (Figure 2) since 1953.

Water flow in the alpine and subalpine zones of the southern Rocky Mountains consists of rapid late spring and summer melting of the snowpack, infiltration of snowmelt into the shallow subsurface, ET losses, and interactions between shallow groundwater and ephemeral snowmelt streams. Early snowmelt infiltrates into materials that generally lack soil ice and are buried in deep snow; channel discharge is low. As melt rates increase in the late spring and early summer, infiltration increases, the soil beneath the snow saturates locally in topographically-favored locations, and the surface flow may rapidly transport water to channels and out of headwater catchments [30]. During later stages of melt, subsurface flow dominates. The flow in the channels that drain the alpine areas peaks in early July on Niwot Ridge, fed mainly by shallow groundwater and local surface runoff beneath snowpacks. In downstream areas, snowmelt-fed streams infiltrate water through locally permeable beds, nourishing near-channel prisms of groundwater. Summer storm duration is not generally sufficient to produce significant flow in channels, and flow usually ceases before late November and resumes the following May [31].

Glaciated headwater catchments in the western USA have been modeled as watertight basins coated by till that impedes infiltration and groundwater flow and where snowmelt runs overland to streams. Williams et al. [15] demonstrated that this model is incomplete and that extensive subsurface pathways provide significant, ephemeral flow to the channel network [32]. This study uses hydrologic data and field observations from four small catchments that drain areas of Niwot Ridge mainly underlain by periglacial deposits (Figure 2): Martinelli (0.26 km$^2$), Saddle (0.26 km$^2$), Como Creek (5.36 km$^2$), and upper Fourmile (1.3 km$^2$), as well as Green Lakes 4 (GL4; Figure 2), an adjacent, marginally glacierized catchment developed in bedrock and till. Observations of hydraulic conductivity, soil-water flow, and groundwater levels on Niwot Ridge [14]) show that the water table generally is <8.5 m below the surface and follows a subdued version of the topography. With the exception of a few days of peak snowmelt, flow in the upper Fourmile channel originates at a large spring complex, which contributes between 18 and 60% of the total basin (64 km$^2$) discharge during summer low flow [33]. The annual runoff efficiency is 88% in GL4, and 24% in Como Creek [29], which flows in thick surficial deposits and likely is a losing stream in much of the area below the tree line [33]. Knowles et al. [13] calculated the water balance loss in Como Creek as being 10.4% of the mean annual precipitation (MAP) after accounting for ET and found that aggregated alpine ET represents 66% of precipitation. After snowmelt slows, the specific yield is extremely low for the Martinelli and Saddle catchments. For example, during a September 2013 rain event that produced substantial flooding throughout the Front Range, the specific yield was 7.8% and 12.4%, respectively, in the Martinelli and Saddle catchments, compared to 60.8% yield from GL4 (D. Dethier, unpublished data).

We hypothesize that the relatively high hydraulic conductivity and subsurface architecture of periglacial deposits in the landscapes adjacent to glaciated valleys control their hydrologic properties. Previous analyses of the shallow subsurface on Niwot Ridge emphasized soil moisture [13], mixing of water from different source areas [30], and groundwater flow and hydraulic conductivity near Saddle stream [14,34]. This detailed, local research focused on the relationship of alpine soil hydraulic conductivity to soil and deeper geologic materials using distributed field measurements and contributed to an understanding of flow in the shallow critical zone. We use 130 measurements of hydraulic conductivity from transects that sample alpine and subalpine soils on Niwot Ridge along a gradient from stations D1 (3739 m) to C1 (3022 m). We combine these data with soil moisture values along the transects and hydraulic conductivity tests performed at the C1 wells to help characterize the response of soil water and the shallow aquifer to annual spring snowmelt. We also hypothesize that the "leaky" nature of periglacial deposits and shallow subsurface storage [14,33] means that coupled infiltration and subsurface flow accommodate peak snowmelt rates in most alpine areas and that seasonal snowmelt in upland catchments contributes substantially to local groundwater recharge through channel beds in downstream areas (see [35]).

## 3. Methods

Soils mapped and classified by the USDA-NRCS (https://websoilsurvey.sc.egov.usda.gov/App/WebSoilSurvey.aspx; accessed on 18 December 2016) [16] provided a focus for field studies along 13 representative 30-m transects that ranged from climate monitoring site C1 (elevation 3022 m) west to site D1 (elevation 3739 m) on Niwot Ridge (Figure 2). The distributed test sites included eight different soil types mapped by Burns [20]; transect selection also was guided by parent material, elevation, and local topography. Transect measurements included hydraulic conductivity and soil volumetric moisture centered at depths of 5 cm (6 cm effective thickness) and 10 cm (13 cm effective thickness), collected at 100 cm intervals using Decagon Device's 5TM and 10HS Soil Moisture Probes (https://www.metergroup.com/environment/products; accessed on 1 July 2022), which measure dielectric permittivity. During an unusual two-week period without precipitation, we were able to remeasure three transect sites (COMO1, COMO3, and COMO5) to determine the change in volumetric water content over time and thus a field ET rate.

Soil infiltration experiments were conducted at a spacing of 3 m after removing the O-horizon, using a Decagon Mini Disk Infiltrometer (Model S) (https://www.metergroup.com/environment/products/mini-disk-infiltrometer/ accessed on 1 July 2022). We calculated the near-saturated hydraulic conductivity from the soil infiltration measurements using [36] and the suggestions of Dohnal et al. [37]. In this paper, we use the simpler expression soil "hydraulic conductivity" instead of "near-saturated hydraulic conductivity". Composite samples of the mineral soil were collected from each transect for settling-column analysis of texture in the laboratory. In order to account for high but unmeasured infiltration rates in alpine areas where open-work boulders (blockfields) form the surface material, we applied Envi 5.0$^{TM}$ supervised classification techniques to September 2004 images collected by the National High-Altitude Photography Program through their EarthExplorer website (earthexplorer.usgs.gov; accessed on 15 January 2017). We classified areas mainly mapped as the Lithic Cryorthents and Rubble land complex of the NRCS (mostly the Minimal Snow Cover Soil Complex (Patterned Ground) of [20]) between the tree line (~3400 m) and the crest of Niwot Ridge. We checked the classification locally using field traverses of two test areas that totaled 15,000 m$^2$.

At 10 previously untested observation wells developed in till at the subalpine C1 area (Figure 2), we measured hydraulic conductivity (K) using falling head slug techniques, a HOBOware (Onset HOBO and InTemp Data Loggers; https://www.onsetcomp.com; accessed on 1 July 2022) pressure transducer that recorded at 1 s intervals for 30 min and, for one well that displayed extremely low values, hand measurements for an additional 2 to 16 hr. Slug tests measure the hydraulic conductivity of a relatively small volume of material adjacent to the open (screened) portion of the well over a relatively short time period. Geologic logs of the C1 wells suggest that they developed in relatively uniform non-stratified, nonsorted till, somewhat weathered in the upper meter. We assume that the small volume sampled by the slug tests in 10 wells fairly represents the relatively uniform subsurface material, as suggested by King [14] from her slug tests at wells near SADL1(Figure 2). We analyzed the data using Fetter's [38] simplification of the Hvorslev method [39] and compared these results to those from the Shapiro–Greene [40] and Bouwer–Rice [41] methods using U.S. Geological Survey algorithms (water.usgs.gov; accessed on 15 January 2017). Hvorslev calculated hydraulic conductivity (K; Equation (2)) as:

$$K = \frac{r_w^2 \ln(L_e/r_c)}{2L_e t_{37}} \tag{2}$$

$K$ = Hydraulic conductivity (m·s$^{-1}$)
$r_w$ = Radius of the well screen (m)
$Le$ = Length of the well casing (m)
$r_c$ = Radius of the well casing (m)
$t_{37}$ = Time it takes for the water level to reach

37% of its initial height at $t_0$ (s)

In our analyses, we also drew on the extraordinary wealth of monitoring data and analysis, both published and unpublished, available for Niwot Ridge and adjacent areas from the Niwot Ridge LTER database (https://nwt.lternet.edu/data-catalog; accessed on 15 January 2017). Niwot SNOTEL data are from http://www.wcc.nrcs.usda.gov/nwcc/site?sitenum=663&state=co; accessed on 1 November 2016).

## 4. Results

Field measurements allowed us to quantify how rapidly meltwater infiltrates and moves through the partially saturated zone, ambient soil volumetric moisture content, and hydraulic conductivity in an alpine and subalpine critical zone dominated by a short season of snowmelt.

### 4.1. Hydraulic Conductivity Measurements

Infiltration measurements at sites from the rocky alpine at 3700 m to subalpine forested at 3045 m gave hydraulic conductivity ranging from ~$5 \times 10^{-4}$ to <$10^{-7}$ m·s$^{-1}$ (Figure 4). The soils are gravel-rich loamy sand with a fines (fine silt + clay) content that ranged from 3.7 to 13.5%. Hydraulic conductivity was inversely correlated with the fines content of soil samples ($r^2 = 0.23$; $p < 0.001$) but not correlated with organic matter content. Four sites, at mainly lower elevations, included slightly finer subalpine forest soils developed on weathered till and gave lower K values. Nine sites were in soils developed from periglacial deposits in subalpine and alpine areas and gave higher K values, including two sites (COMO 1 and COMO 5) previously mapped by the NRCS as the Leighcan family (till substratum). We assume that infiltration measurements made near and below the tree line are representative, but measured values from the highest-elevation sites had to be generalized by using estimates for the area of openwork boulders where rates could not be measured directly.

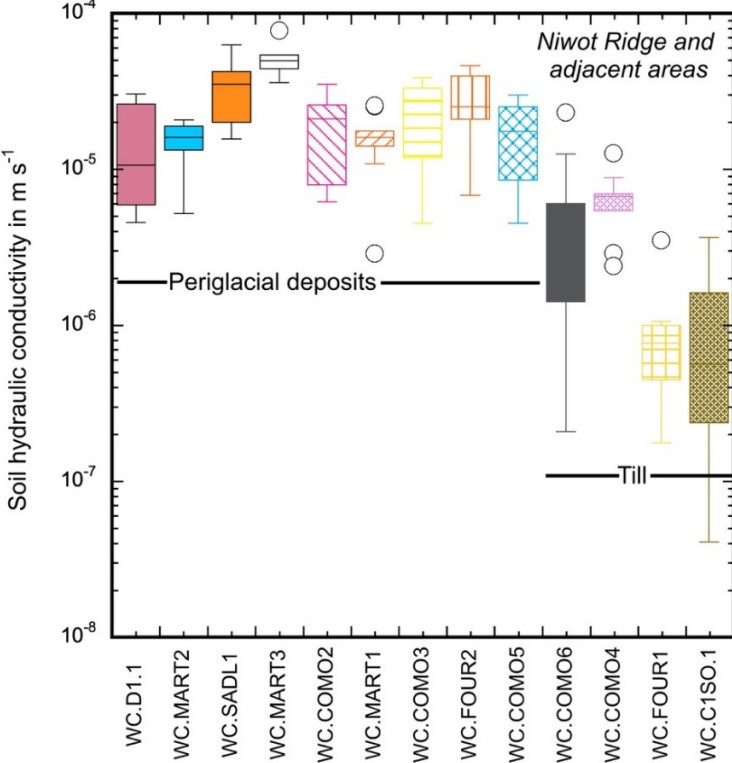

**Figure 4.** Box-and-whisker plot of measured hydraulic conductivity, Niwot Ridge. Transects are ordered from highest (D1.1) to lowest elevation (C1SO.1). Each site includes 10 measurements; 6 extremely slow rates were not included in the analysis. White circles mark outliers.

Field transects suggested that openwork boulders locally covered 30 to 46% at two sites with patterned ground, one at 3700 m and the other at 3500 m. Digital supervised classification of the entire area separated the alpine portion of Niwot Ridge into three zones based on elevation and gave an estimate of 23.0% patterned ground over 3600 m, 2.7% patterned ground between 3600 m and 3500 m, and 5.6% between 3500 m and the tree line. We used the supervised classification values and assumed that openwork boulders have a hydraulic conductivity of $1.2 \times 10^{-2}$ m·s$^{-1}$, the average measured on nearby blockfields containing both coarse debris and fines [42] and comparable to values reported from alpine talus by [25]. We generalized each measured alpine hydraulic conductivity using the image classification percentages and area/elevation data (Figure 5). The hydraulic conductivity values calculated by including the exposed area of openwork boulders show that the alpine areas support higher rates than the adjacent subalpine areas. If high but unmeasured rates in the alpine talus and other cirque deposits (CQ; Figure 2) were included, the infiltration gradient from alpine to subalpine areas would be even larger.

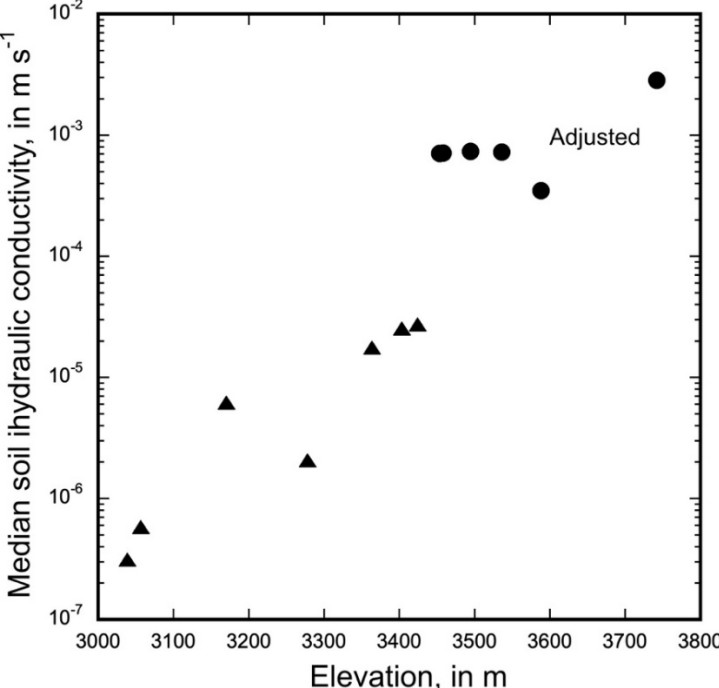

**Figure 5.** Elevation plot of median hydraulic conductivity on Niwot Ridge. Subalpine values shown as triangles. Six measured alpine values (circles) were adjusted upward using the infiltration rate ($1.23 \times 10^{-2}$ m s$^{-1}$) of Davinroy [42], based on the percent of openwork gravel estimated using image classification (23.0% above 3600 m, 2.7% between 3600 m and 3500 m and 5.6% between 3500 m and tree line). Upper and lower quartile boundaries are displayed in Figure 4.

*4.2. Soil Moisture Content and Flow*

At depths of 5 and 10 cm, volumetric soil moisture content ranged from about 45% adjacent to a recently melted snowdrift to <5% in subalpine soils where snowmelt occurred >2 months before sampling. The moisture data (Supplementary Material) reflect a range of local site conditions in mid-July, soil drainage after snowmelt, June rainfall events, and the influence of ET. Data from the lysimeter array at site C1 (Figure 6) illustrate how the seasonal pattern of volumetric soil moisture evolved in till-derived soils from the near-surface to deep in the parent material as the wetting front migrated downward to recharge the water table in 2016.

Wetting front moisture at site C1 rose from low values to a peak in early April in the shallow soil and in middle to late May at depths of 70 to 150 cm, decaying rapidly in June and July. Summer rainfall events mainly influenced the shallow soil. Wetting-front

velocities measured at 10, 20, 30, 50, 70, and 150 cm at the saturated/unsaturated interface had mean values of ~$8.4 \times 10^{-7}$ m s$^{-1}$ during the peak melt period with a maximum of $2.5 \times 10^{-6}$ m s$^{-1}$ at a depth of 30–50 cm. Soil moisture loss from the 5 to 30 cm depth interval averaged ~1 mm d$^{-1}$ during mid-July 2016.

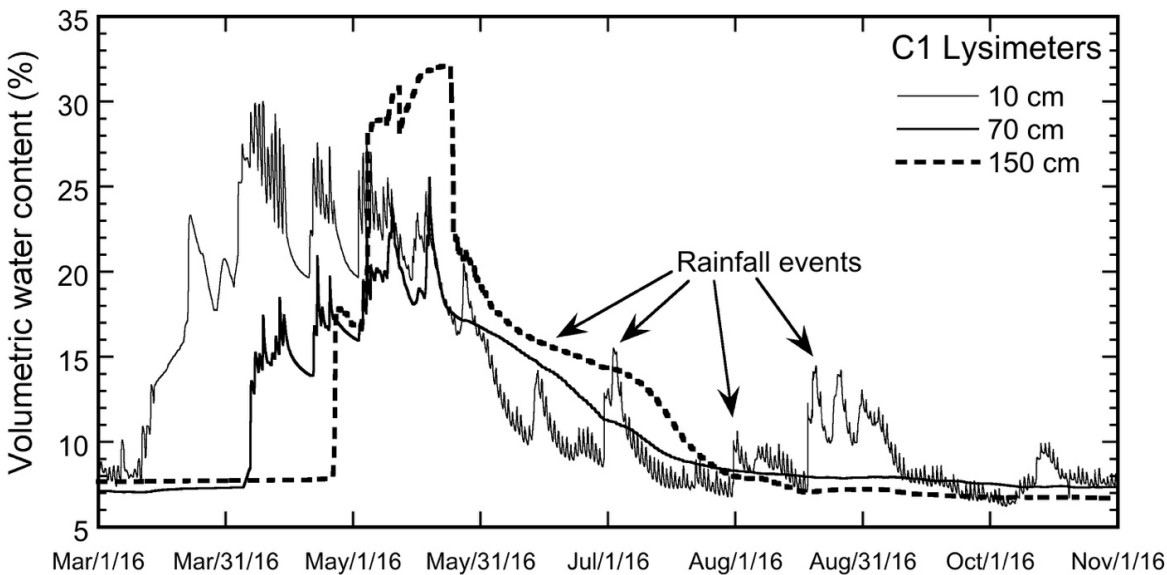

**Figure 6.** Seasonal volumetric soil moisture change measured at 10-min intervals at depths of 10, 70 and 150 cm below the surface at site C1 in 2016. Data from (colorado.edu/mrs/; accessed on 12 October 2016).

The two-week period without precipitation permitted the calculation of a field-based upper limit for ET rates during midsummer at a group of alpine meadow sites (Figure 7).

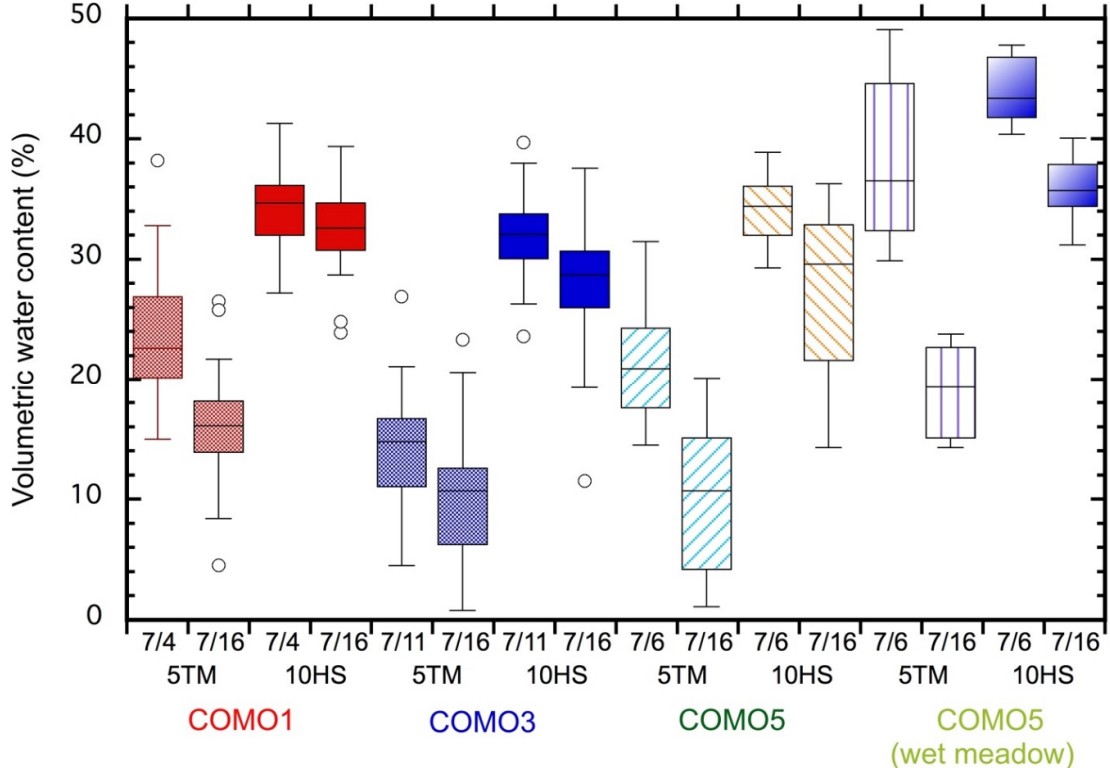

**Figure 7.** Box-and-whisker visualization of repeated soil moisture transects at three Niwot Ridge

sites showing overall decreases in volumetric water content at both 5- and 10-cm depths over periods of 12, 5, and 10 days for COMO1, COMO3, and COMO5, respectively (Supplementary Material). Transects COMO1 and COMO3 were 30 m long and included 31 measurements at each depth. COMO5 was divided into a 25-point and a 6-point section where the transect crossed from a meadow to a wet meadow. Circles mark outliers.

During a sunny, 5-to-12-day period when the mean temperature was 14.7 °C, the calculated values ranged from ~1 mm d$^{-1}$ (dry meadow) to 6 mm d$^{-1}$ (wet meadow) at 5 cm depth and 0.5 mm d$^{-1}$ (dry meadow) to 3 mm d$^{-1}$ (wet meadow) at 10 cm depth. Losses by ET generally were greater from the shallow mineral soil.

### 4.3. Hydraulic Conductivity and Groundwater Flow at the C1 Well Field

The calculated saturated hydraulic conductivity (K) values varied widely at the C1 wells, which are sited in ~140 kyr till at least 30 m thick. The median values calculated from falling head slug tests using different analytical [38] gave ~23 × 10$^{-6}$ m s$^{-1}$ (Bouwer–Rice), ~5.0 × 10$^{-6}$ m s$^{-1}$ (Hvorslev), and ~1.1 × 10$^{-6}$ m s$^{-1}$ (Shapiro–Green). The Hvorslev and Shapiro–Green values were correlated ($p < 0.001$) but not significantly different, whereas the Shapiro–Green values were significantly lower than Bouwer–Rice ($p < 0.0007$), but there was no correlation between values calculated by the two techniques. The calculated values for individual well tests varied by 1 to 2 orders of magnitude across the three techniques; most repeat slug tests produced values within 2× of each other (Table 1) for individual analytical techniques. We did not note any consistent spatial differences in hydraulic conductivity for the wells we sampled. The deepest well (C1-SW.2) and a nearby well (-SW.1) give among the lowest values using Hvorslev, but not the other calculation methods.

**Table 1.** Summary of saturated hydraulic conductivity (K) values calculated by several methods from falling head slug tests at the C1 well field, Niwot Ridge, Colorado.

| Well | Date Tested | Total Test Time (s) | Hvorslev K (m·s$^{-1}$) * | Shapiro-Greene K (m·s$^{-1}$) ^ | Bouwer-Rice K (m·s$^{-1}$) ° | 20% Linear K (m·s$^{-1}$) # |
|---|---|---|---|---|---|---|
| C1.BEL | 7/18/16 | 13 | 8.84 × 10$^{-4}$ | 1.09 × 10$^{-5}$ | 4.16 × 10$^{-5}$ | 5.00 × 10$^{-4}$ |
| C1.SW.1 | 7/17/16 | 12,720 | 4.86 × 10$^{-7}$ | 4.76 × 10$^{-7}$ | 4.37 × 10$^{-5}$ | 2.06 × 10$^{-4}$ |
| C1.SW.1 | 7/18/16 | 9529 | 5.14 × 10$^{-7}$ | 5.24 × 10$^{-7}$ | 4.37 × 10$^{-5}$ | 2.78 × 10$^{-4}$ |
| C1.SW.2 | 7/16/16 | 1470 | 4.71 × 10$^{-6}$ | 4.23 × 10$^{-6}$ | 3.23 × 10$^{-6}$ | 1.04 × 10$^{-3}$ |
| C1.SW.2 | 7/18/16 | 1486 | 4.87 × 10$^{-6}$ | 4.23 × 10$^{-6}$ | 3.23 × 10$^{-6}$ | 1.45 × 10$^{-4}$ |
| C1.SO.1 | 7/16/16 | 1448 | 1.77 × 10$^{-4}$ | 3.42 × 10$^{-6}$ | 4.64 × 10$^{-5}$ | 1.34 × 10$^{-5}$ |
| C1.SO.2 | 7/14/16 | 1399 | 1.75 × 10$^{-5}$ | 2.69 × 10$^{-6}$ | 8.98 × 10$^{-6}$ | 2.36 × 10$^{-5}$ |
| C1.SO.2 | 7/16/16 | 1452 | 3.12 × 10$^{-5}$ | 2.73 × 10$^{-6}$ | 8.97 × 10$^{-6}$ | 2.46 × 10$^{-5}$ |
| C1.SO.3 | 7/16/16 | 1434 | 8.41 × 10$^{-7}$ | 1.04 × 10$^{-6}$ | 8.28 × 10$^{-6}$ | 6.05 × 10$^{-4}$ |
| C1.SO.3 | 7/18/16 | 4010 | 5.91 × 10$^{-7}$ | 7.31 × 10$^{-7}$ | 8.28 × 10$^{-6}$ | 3.36 × 10$^{-4}$ |
| C1.SO.4 | 7/16/16 | 1411 | 1.12 × 10$^{-5}$ | 1.17 × 10$^{-6}$ | 8.68 × 10$^{-6}$ | 4.49 × 10$^{-5}$ |
| C1.ST.1 | 7/17/16 | 1414 | 5.19 × 10$^{-6}$ | 5.19 × 10$^{-7}$ | 4.64 × 10$^{-5}$ | 1.06 × 10$^{-3}$ |
| C1.ST.3 | 7/17/16 | 1445 | 1.40 × 10$^{-5}$ | 6.67 × 10$^{-7}$ | 4.60 × 10$^{-5}$ | 1.36 × 10$^{-4}$ |
| C1.ST.4 | 7/18/16 | 1434 | 3.08 × 10$^{-6}$ | 7.92 × 10$^{-7}$ | 3.79 × 10$^{-5}$ | 5.53 × 10$^{-5}$ |
| Mean | | | 8.25 × 10$^{-5}$ | 2.44 × 10$^{-6}$ | 2.54 × 10$^{-5}$ | 3.19 × 10$^{-4}$ |
| Std.Dev. (mean) | | | 2.35 × 10$^{-4}$ | 2.81 × 10$^{-6}$ | 1.92 × 10$^{-5}$ | 3.58 × 10$^{-4}$ |
| Median | | | 5.03 × 10$^{-6}$ | 1.10 × 10$^{-6}$ | 2.34 × 10$^{-5}$ | 1.75 × 10$^{-4}$ |
| 95% C.I. (median) | | | 8.41 × 10$^{-7}$, 1.75 × 10$^{-5}$ | 6.67 × 10$^{-7}$, 3.42 × 10$^{-6}$ | 8.28 × 10$^{-6}$, 4.37 × 10$^{-5}$ | 4.49 × 10$^{-5}$, 5.0 × 10$^{-4}$ |

* Calculated using Fetter's [38] simplification of Hvorslev [39]; ^ Calculated using AQTESTSS (https://pubs.usgs.gov/of/2002/ofr02197) implementation of [40]; ° Calculated using AQTESTSS (https://pubs.usgs.gov/of/2002/ofr02197) implementation of [41]; # Uses a linear approximation to the final 20% (by time) of the falling head curve. Provides an upper limit.

Monitoring of groundwater levels by personnel from the University of Colorado Mountain Research Station provided additional observations about groundwater flow at C1. Daily records of water level at well C1.ST.1, located about 50 m from Como Creek, demonstrated an annual 4 to 5 m fluctuation of groundwater levels from peak values on ~1 July, followed by a slow exponential decay to the lowest values before snowmelt began in late spring (Figure 8) from 2012 to 2016. Well C1.SW.2, some 75 m from Como Creek, displayed a similar temporal pattern of fluctuations but amplitudes were ~1.5 m. Long-term measurements at the C1 Snotel site show that all local snow typically melts by 30 May (http://www.wcc.nrcs.usda.gov/nwcc/site?sitenum=663&state=co; accessed on 1 November 2016); peak groundwater levels are delayed until ~1 July, soon after peak flow in Como Creek and other drainages that head in the late melting alpine snowfields of Niwot Ridge.

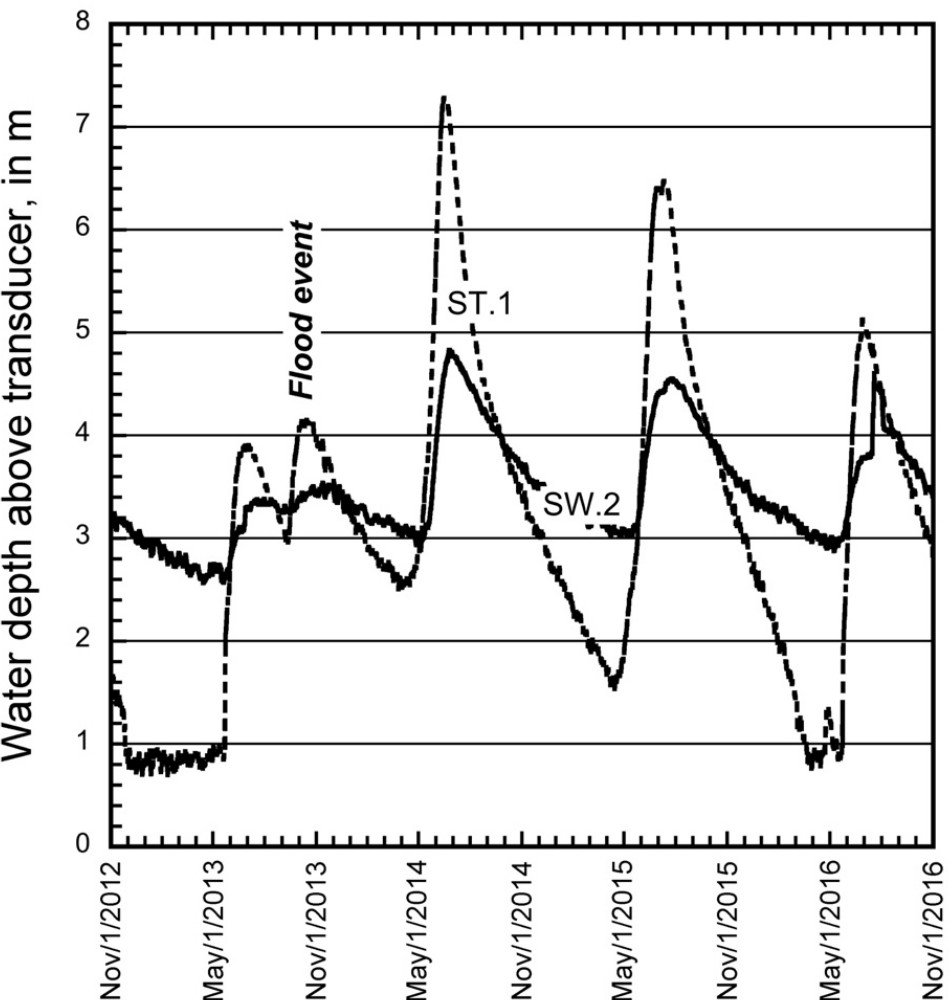

**Figure 8.** Daily water levels at C1 wells ST.1 and SW.2 in 2012–2016 (https://nwt.lternet.edu/data-catalog; accessed on 10 December 2016). Snowmelt in 2013 was unusually low, 2014 was high and 2015 and 2016 were near average. Peaks in 2016 were perturbed by well tests from this study.

Monthly measurements of water levels at other wells in the C1 field (https://nwt.lternet.edu/data-catalog; accessed on 10 December 2016) showed that the water table was ~6 m below Como Creek before snowmelt began (Figure 9) in 2016 and rose to the surface coincident with increased discharge in mid-June. The water table rose at least 4 m as far as 40 m from Como Creek and remained high in mid-July. Water levels dropped in late summer and fall.

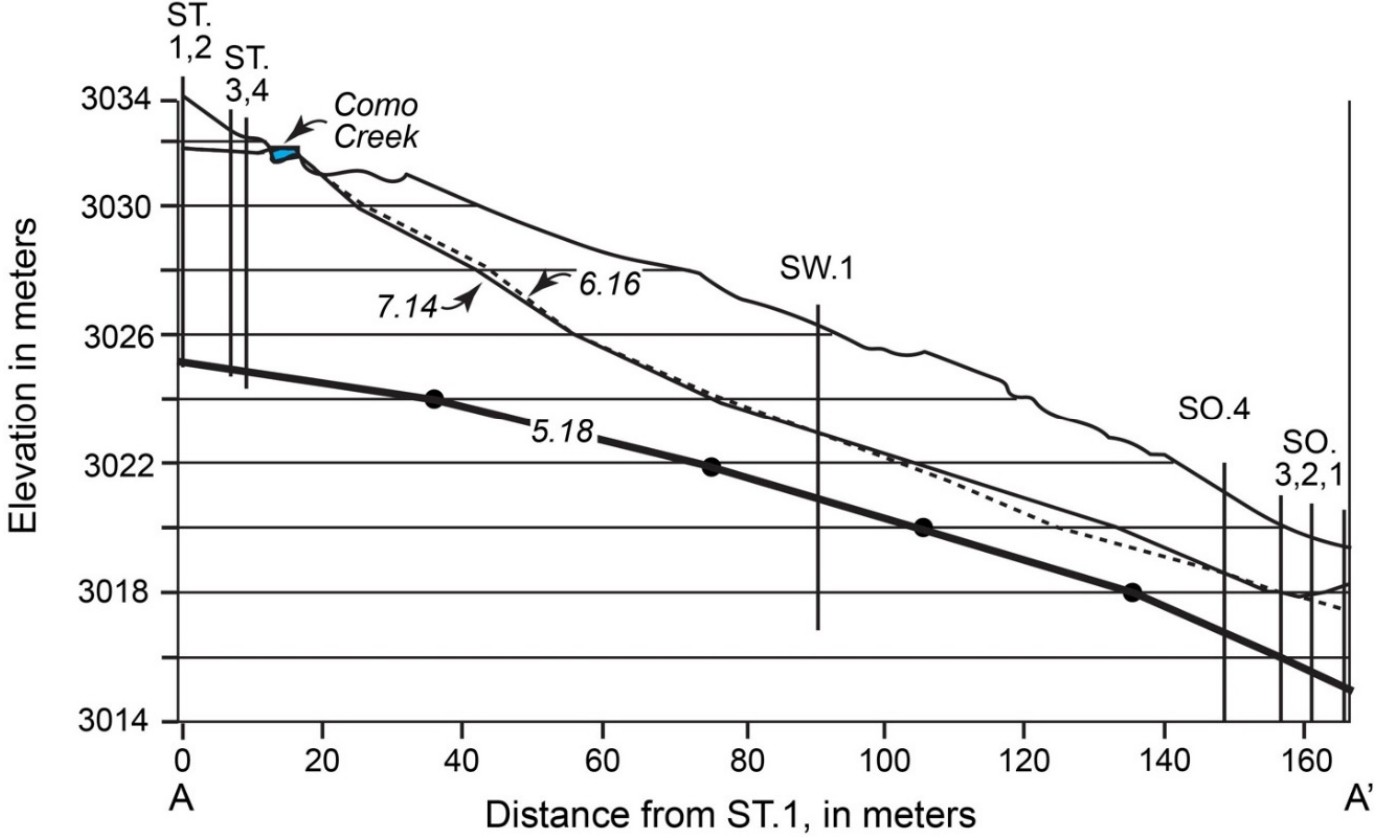

**Figure 9.** Section showing water table elevations on 5.18, 6.16 and 7.14.16 along a flow line normal to contours (see Figure 2 inset). Position of Como Creek portrayed by projecting its location 15 m north; C1 well locations projected 10 to 50 m to the section, parallel to lines of equal head calculated by spatial interpolation (kriging) of data from all 10 wells, Como Creek, and its flood channels.

## 5. Discussion

Mountainous areas serve as critical and increasingly vulnerable sources of water for downstream areas during the summer months. Winter and early spring snowfall on Niwot Ridge, Colorado, and in similar alpine and subalpine settings (e.g., [3,43]) infiltrates the soil as snowmelt mainly during a short period in late spring and early summer and percolates through the soil and in surficial materials to the water table. Shallow groundwater then travels downgradient to intersect surface channels that transport snowmelt to lower, drier parts of catchments. On Niwot Ridge and likely in other high-elevation mountainous areas, hydraulic conductivities are higher in the alpine and upper subalpine zone, where soils are derived from periglacial deposits, and generally lower where the critical zone is developed in till or weathered bedrock at lower elevations (Figure 10). As summer temperatures and ET increase and snowmelt occurs earlier, alpine soils are likely to become more vulnerable to drought, and groundwater levels in the critical zone may lower, affecting the connectivity between late-melting snow, flow in meltwater streams, and the areas they affect downstream [3].

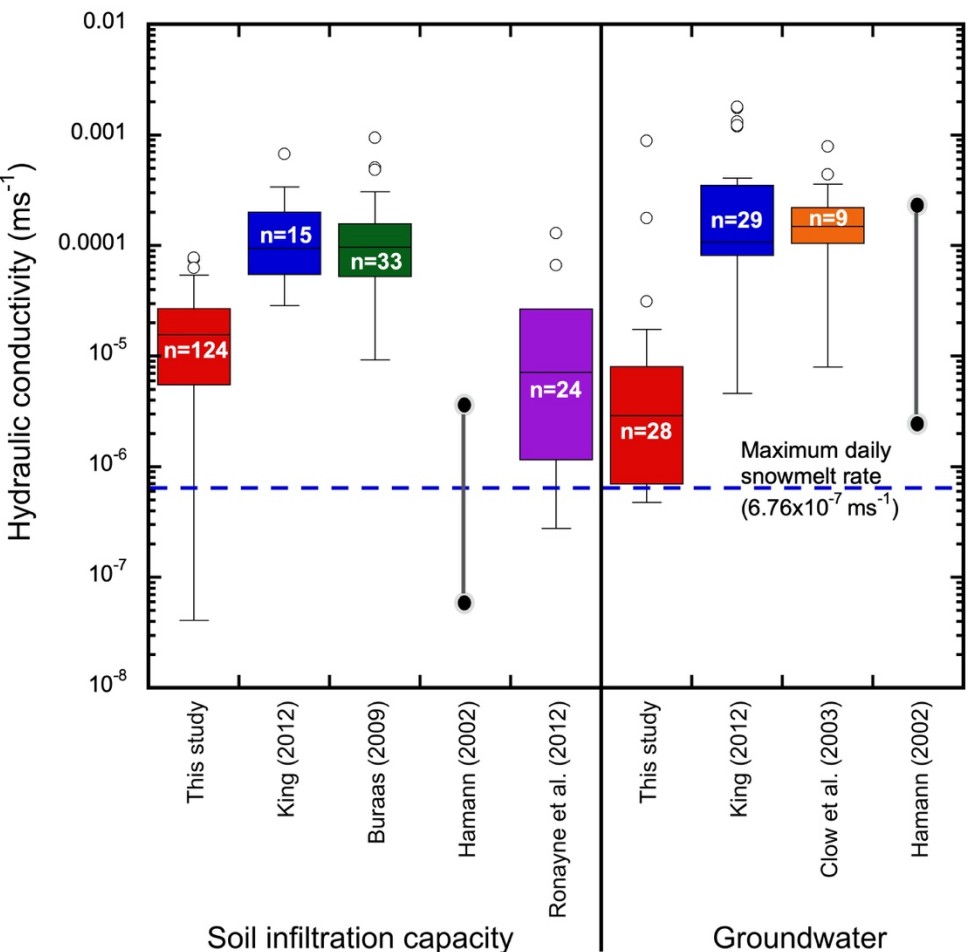

**Figure 10.** Summary statistics for saturated hydraulic conductivity (K) values from this study and other studies from Niwot Ridge and adjacent areas (see also [44]). Six extremely low soil hydraulic conductivities from this study were not plotted. The 28 groundwater values from this study plot the range in subalpine till calculated using the Hvorslev and Shapiro–Greene methods. Hamann [45] reported a range of values. The dashed line is the maximum daily snowmelt rate recorded at the C1 Snotel site from 1981 to 2016 (http://www.wcc.nrcs.usda.gov/nwcc/site?sitenum=663&state=co; accessed on 1 November 2016).

Measurements from this research and from other studies in nearby mountain areas demonstrate the range of saturated hydraulic conductivity values in soils, periglacial deposits, till, and near-surface fractured bedrock derived from granitic and high-grade metamorphic rocks. Values calculated using the Bouwer-Rice analysis of the same slug-test data at C1 are an order of magnitude higher because the analysis is based on a different portion of the recovery curve [41]. Future pumping tests might help clarify which analytical method is appropriate for the site. Soil hydraulic conductivity and groundwater K values are comparable. Rates in periglacial deposits, sandy regolith [46], and latest Pleistocene till derived from granite [47] are significantly higher than those measured in older till derived from metamorphic rocks (this study). Measured hydraulic conductivities indicate that most alpine snowmelt will infiltrate except where local topography or shallow, unfractured bedrock drives convergent flow or local ponding. Combined with previous geophysical studies focused on the architecture of the alpine critical zone [21] and measurements from other N. American alpine areas [48] infiltration capacities reported here support the findings in [15] that alpine and subalpine snowmelt infiltrates in most areas and that only some catchments act as 'teflon basins' that contribute water to streams primarily as surface runoff.

Hydrologic [33,49] and geochemical studies [32,34,50] show that surface and subsurface flow, driven by infiltrating snowmelt, persists for weeks to a few months in alpine areas. In lower-elevation subalpine and upland zones, where precipitation amounts are lower, and snowmelt occurs earlier, streams fed by alpine snowmelt transition from "gaining" to "losing", and a significant proportion of flow infiltrates through stream beds [2,51], locally recharging a near-channel groundwater prism (Figure 9) and contributing to deeper groundwater flow. At Niwot Ridge, soil water in the spring at site C1 closely tracks the melting snowpack, whereas groundwater is at its highest levels in early summer (Figures 6 and 8), coincident with peak alpine melt and streamflow. Groundwater levels in wells at C1 reflect infiltration from Como Creek and its flood channels during the peak melt season in 2016 (Figure 9). Slug tests in the wells at C1 gave low K values (Table 1) for the subalpine till in screened zones, but groundwater levels responded rapidly to the seasonal snowmelt pulse (Figures 8 and 9), indicating that streamflow infiltrated through fluvial deposits or more permeable near-surface till. Comparable streambed infiltration and connectivity produced reaches where stream channels were dry downstream from late-melting snowdrifts in the Martinelli, Saddle, and upper Fourmile catchments [3]. Our research does not demonstrate that streambed infiltration by snowmelt extends downstream from the areas that we have studied, but the literature on "losing" or "disconnected" streams suggests that such streambed infiltration losses are probable (e.g., [52–54]) and help control processes in the downstream hyporheic zone, including the diurnal pattern of discharge, measured downstream in snowmelt-charged streams, such as Como Creek [53].

The warming climate in mountainous areas will likely make both alpine and downstream areas more vulnerable by decreasing the amount and duration of early summer runoff [8] and its timing, as well as by lowering moisture levels in alpine and subalpine soils. After snowmelt ceases, ET rates mainly control late summer water flux from Niwot Ridge, Colorado. Measured short-term rates at three sites showed rapid decreases in water content in upper subalpine and alpine meadow soils, highlighting the sensitivity of near-surface storage to short periods of drought. Water loss from several decimeters deep in subalpine till soils at C1 averaged ~1 mm d$^{-1}$ during the same period. Soil water losses during a warm, dry period were correlated with the initial volumetric value and are comparable to values suggested by the modeling and results of Knowles et al. [13] for alpine soils. Surface runoff after general snowmelt ceases is unusual and occurs until midsummer, at present, only where moist meadow soils associated with persistent snowdrifts connect with channels [20,55], but these zones of storage and connectivity are likely to decrease in the future.

The data presented here contribute to ongoing hydrological research by predicting probable values for hydraulic conductivity and saturated hydraulic conductivity in the snowmelt-producing alpine and subalpine zone of mountain areas. Our data support the hypothesis that the hydraulic conductivity on Niwot Ridge, Colorado, and in similar alpine and subalpine settings in North America mainly exceeds maximum snowmelt rates. In changing climatic regimes more prone to early snowmelt and summer warmth, snowmelt infiltration and soil moisture may decline in headward catchments, with implications for the timing and amount of water that flows downstream in the catchment [2,6]. The environmental significance of infiltration through the beds of "losing" snowmelt streams remains an important research topic.

## 6. Conclusions

Most snowmelt in the alpine and upper subalpine critical zone of Niwot Ridge (USA) flows rapidly through the shallow subsurface in layered periglacial deposits to a network of ephemeral streams, as suggested by Williams et al. [15]. Direct infiltration of snowmelt and local infiltrated water from stream channels through weathered till feeds groundwater in the subalpine critical zone. Hydraulic conductivity values of $5.6 \times 10^{-7}$ to $4.96 \times 10^{-5}$ m s$^{-1}$ exceed maximum rates of snowmelt in most areas. Transfer of snowmelt to groundwater is mediated by the position of late-melting snow patches and relatively high rates of

evapotranspiration (ET) from moist soils during warm summer periods and remains significant ($\sim$1 mm d$^{-1}$) in subalpine tills at 30 cm depth. Water that originates in alpine and subalpine areas contributes substantially to flow in and beneath channels in lower portions of catchments long after snowmelt has ceased. In changing climatic regimes more prone to early snowmelt and summer warmth, snowmelt infiltration and soil moisture may decline in mountainous areas, making the shallow critical zone more vulnerable to drought and changing the timing and amount of water that flows downstream in the catchment.

**Supplementary Materials:** The following supporting information can be downloaded at: https://www.mdpi.com/article/10.3390/w14152317/s1.

**Author Contributions:** Conceptualization, N.W. and D.P.D.; methodology, N.F and J.F.F.; formal analysis, N.W.; data curation, D.P.D.; writing—original draft preparation, N.W. and D.P.D.; writing—review and editing, D.P.D. and J.F.F.; All authors have read and agreed to the published version of the manuscript.

**Funding:** This research was funded by the Keck Geology Consortium (NSF EAR-1062720).

**Institutional Review Board Statement:** Not applicable.

**Informed Consent Statement:** Not applicable.

**Data Availability Statement:** The data presented in this study are available on request from the corresponding author.

**Acknowledgments:** We are grateful for the logistical help provided by personnel from the University of Colorado Mountain Research Station and unpublished data for this research provided by the Niwot Ridge LTER program (NSF DEB–1637686). Field studies and measurements in the Niwot Ridge area were performed in cooperation with the NSF-sponsored NWT Long-Term Ecological Research (LTER) project. Technical and editorial comments from three anonymous reviewers and the editors greatly improved an earlier version of this paper.

**Conflicts of Interest:** The authors declare no conflict of interest.

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
