# Peer review of "Snowmelt-Driven Seasonal Infiltration and Flow in the Upper Critical Zone, Niwot Ridge (Colorado), USA"

_water, doi:10.3390/w14152317_

Round 1

Reviewer 1 Report

Dear Editor, è

Thank you for the opportunity to review this manuscript. In their work, the authors focus on the snowmelt-driven seasonal infiltration and flow in the upper critical zone in the Niwot Ridge area, which is one of the most studied alpine sites in North America. The authors characterize the permeability an infiltration capacity of oil in the alpine and subalpine domains of the study zone.

General comments.

The reviewer finds the manuscript interesting, and the topic will sufficiently engage WATER's wide readership. The experiment setting and applied methods are fine, and the obtained results are in line with similar results obtained in other areas. In this line, I miss in the introduction a state of the art regarding the hydrological characterization of alpine zones, which is a very active research field. In this vein, I suggest the authors to review, among others, the works of Roy and Hayashi (2009), McClymont et al. (2011), Langston et al. (2011), Muir et al. (2011), Kurylyk and Hayashi (2017), Christensen et al. (2020), Hayashi (2020), and references therein. Besides, although the discussion section is fine, comparing your results with those obtained by the aforementioned authors would enrich largely this work

Particular comments:

Fig.1: In this figure, it seems that Qsw is stream water when stream water is actually the sum of Qsw and Qgw. I would mark Qsw as an arrow coming from the dashed blue line which is located downgradient from the snow zone. Besides, the green shaded area corresponding to Lithic Cryorthents overlaps the name labels of some sample sites thus hindering the reading of such labels. Please correct.

Fig. 2: Despite being the Colorado State so well known by almost everyone, here it is missing a larger-scale situation map to allow the clueless readers to locate correctly the study zone.

Line 14: I assume that the infiltration capacity has dimensions of [LT-1] and therefore “ms-1” means m/s. Nevertheless, a non-expert reader might understand that “ms-1” corresponds to the unit of a variable with dimension [T-1] given that “ms” means milliseconds in the International System of units. In other words, the unit notation “ms-1” is confusing. To avoid this issue, you should replace”ms-1” with “m/s” throughout the manuscript.

Lines 40-44: From a general standpoint, all these variables (i.e. P, E, ET, Q, and DS) are defined as flux variables, and therefore, they have associated a dimension of [LT-1]. As a result, and being consistent with the Author’s unit scheme, one would expect these variables to show mm/d as a unit. Please modify the text accordingly.

Line 126: After D1 please add (Fig.2).

Line 157: Where it reads (GL4) it should read (GL4, Fig.1)

Line 157: Please define the acronym MAP

Line 180: At the end of the line I would include the reference of Jódar et al., 2022.

Lines 392 and 419: Please change “mmd-1” by “mm/d”

Bibliography:

Roy, J. W., & Hayashi, M. (2009). Multiple, distinct groundwater flow systems of a single moraine–talus feature in an alpine watershed. Journal of Hydrology, 373(1-2), 139-150.

McClymont, A. F., Roy, J. W., Hayashi, M., Bentley, L. R., Maurer, H., & Langston, G. (2011). Investigating groundwater flow paths within proglacial moraine using multiple geophysical methods. Journal of Hydrology, 399(1-2), 57-69.

Langston, G., Bentley, L. R., Hayashi, M., McClymont, A., & Pidlisecky, A. (2011). Internal structure and hydrological functions of an alpine proglacial moraine. Hydrological processes, 25(19), 2967-2982.

Muir, D. L., Hayashi, M., & McClymont, A. F. (2011). Hydrological storage and transmission characteristics of an alpine talus. Hydrological Processes, 25(19), 2954-2966.

Kurylyk, B. L., & Hayashi, M. (2017). Inferring hydraulic properties of alpine aquifers from the propagation of diurnal snowmelt signals. Water Resources Research, 53(5), 4271-4285.

Hayashi, M. (2020). Alpine hydrogeology: The critical role of groundwater in sourcing the headwaters of the world. Groundwater, 58(4), 498-510.

Christensen, C. W., Hayashi, M., & Bentley, L. R. (2020). Hydrogeological characterization of an alpine aquifer system in the Canadian Rocky Mountains. Hydrogeology journal, 28(5), 1871-1890.

Jódar, J., Zakaluk, T., González-Ramón, A., Ruiz-Costán, A., Marín-Lechado, C., Martín-Civantos, J.M., Custodio, E., Urrutia, J., Herrera, C., Lambán, L.J., Durán, J.J., Martos-Rosillo, S. (2022). Artificial recharge by means of careo channels versus natural aquifer recharge in a semi-arid, high-mountain watershed (Sierra Nevada, Spain). Science of The Total Environment. https://doi.org/10.1016/j.scitotenv.2022.153937

Reviewer 2 Report

This paper deals with the analysis of snowmelt infiltration and its connectivity to groundwater in downstream areas in Niwot Ridge, Colorado, where infiltration capacity and soil moisture were measured at 13 representative sites along 30 m long transects. The role of snowmelt infiltration in recharging groundwater in a warming climate setting was also discussed.

The novelty of this paper is not clear and the authors failed to identify a clear gap in the literature. The main points of this study are 1 the existence of elevation zonation of infiltration capacity in this area; and 2 streambed infiltration plays an important role to groundwater in downstream areas. Those points are not of interest to the community because they belong in the textbook. The textbook also tells us that infiltration capacity varies with time, is infinitely large at the very beginning in the very dry state and gradually converges to saturated hydraulic conductivity. Therefore, the scientific significance of this paper is limited, although the measured infiltration capacities for periglacial soils are of value.

The analysis of the connectivity between snowmelt infiltration with groundwater table dynamics is not profound enough and is restricted to phenomenal descriptions. I encourage the authors to apply an appropriate hydrological model to understand the mechanisms behind the phenomena presented in this study.

There are also some technical problems, such as in lines 235-236, where the authors claim that infiltration capacity was not correlated with organic matter content. However, in line 195, the organic horizon was removed before measuring infiltration capacity, so I don’t know how the authors can get at this point. But I think that these issues are not so important as the lack of focus, scientific motivation and interest for the scientific research.

For this reason, I regret to suggest to reject the manuscript.

Reviewer 3 Report

General comments

Valid research in the framework of critical zone hydrology in alpine regions in the scope of MPI-Water. However, all the specific comments need to be addressed before publication

Specific comments

Lines 15-16. “Hydraulic conductivities measured by modified Hvorslev technique ranged 15

from 4.86 x10-7 to 1.77 x10-4 ms-1 in subalpine areas”. Sentence un-clear, you need to specify at least the type of deposit

Line 20. “ET”, write evapotranspiration in the abstract

Line 30-31. Add relevant papers on snowmelt-dominated mountain catchments in the western part of the country.

- Medici, G., Engdahl, N. B., & Langman, J. B. (2021). A basin-scale groundwater flow model of the Columbia Plateau regional aquifer system in the Palouse (USA): Insights for aquifer vulnerability assessment. International Journal of Environmental Research, 15(2), 299-312.

- Behrens, D., Langman, J. B., Brooks, E. S., Boll, J., Waynant, K., Moberly, J. G., ... & Dodd, J. W. (2021). Tracing δ18O and δ2H in Source Waters and Recharge Pathways of a Fractured-Basalt and Interbedded-Sediment Aquifer, Columbia River Flood Basalt Province. Geosciences11(10), 400.

Line 39. I suggest explaining the meaning of Critical Zone and the conceptual link of your research with this layer of the geosphere. The definition of Critical Zone is much less popular outside the US

Line 39. Formatting issue

Line 74. State at the end of this paragraph the aim of your research and the 2-3 objectives that you intend to develop.

Lines 104-105. Describe type of igneous and metamorphic rocks that characterized the study site. Granites and gneiss?

Lines 208-216. Briefly discuss the scaling issue underneath the use of slug tests. They don’t represent the Minimum Representative Volume of the aquifer in terms of connectivity of the fracturing network. Not a problem for the geological media (e.g., till) that you investigate??? Please, justify

Lines 299-304. Explain here or above the scaling issue

Lines 299-304. Don’t forget to discuss differences in values between the different methods that you have used for slug tests

Line 299. Any evident spatial difference in terms of hydraulic conductivity? If yes, add one or two sentences

Lines 332-409. Pay more attention in your discussion on the general meaning of your research beyond the study site. Wider implications for alpine regions?

Lines 420. Make clear what you mean by channel, this word in geoscience means many things

Lines 421-425. Expand this topic in your discussion

Line 433. Insert the relevant and recent references suggested above

Figures and tables

Figures 2. Add the USA map

Figure 5. Rotate map to have the depth on the vertical axis

Round 2

Reviewer 2 Report

This manuscript provides field measurements of infiltration capacity and soil moisture along transects in alpine zones above local glacial limits and indicates a downward gradient of infiltration capacity from alpine to subalpine areas. Insightful discussions are also included. In this revision, some terms have been clarified and the quality has been improved. But I am still confused with the definition of soil infiltration capacity.

In hydrology, infiltration capacity is also defined as “the maximum rate of infiltration at the soil/atmosphere interface”, but infiltration capacity decreases as the soil moisture content of soils surface layers increases. Please check out this link for some basic information: https://www.engr.colostate.edu/~ramirez/ce_old/classes/cive322-Ramirez/CE322_Web/InfiltrationComputationsExample.htm

In general, the infiltration capacity has a range of positive infinity to saturated hydraulic conductivity (lower limit). Therefore, please clarify at what time the infiltration capacity is measured in this study.

Author Response

Reviewer 2 is correct that we have been too informal with our use of terms for the maximum rate at which water can enter and flow through the soil. Previously we used “infiltration capacity” to refer to hydraulic conductivity (or “saturated hydraulic conductivity), a term often associated with groundwater flow. To be hydrologically correct and to simplify, we have replaced “infiltration capacity” with “hydraulic conductivity” throughout the paper and figures and have added a note (see Track Changes version) explaining our use of terms. We used our field measurements of infiltration to calculate soil hydraulic conductivity using a method proposed by Zhang (1997) and the modifications suggested by Dohnal et al. (2010).   We appreciate the reviewer’s patience and help, including the calculated examples.